# The Effects of Agricultural Plastic Waste on the Vermicompost Process and Health Status of *Eisenia fetida*

José A. Sáez [1,*], Angie M. Pedraza Torres [2], Zbigniew Emil Blesa Marco [1], Francisco Javier Andreu-Rodríguez [1], Frutos C. Marhuenda-Egea [3], Encarnación Martínez-Sabater [1], María J. López [4], Francisca Suarez-Estrella [4] and Raúl Moral [1]

[1] Center of Research and Innovation in Agri-Food and Agro-Environmental (CIAGRO-UMH), GIAAMA Research Group, University Miguel Hernández, Carretera de Beniel Km 3.2, 03312 Orihuela, Spain
[2] Laboratory of Ecotoxicology, Institute of Environmental Science (ICAM), University of Castilla-La Mancha, Avda. Carlos III, 45071 Toledo, Spain
[3] Department of Agrochemistry and Biochemistry, Multidisciplinary for Environmental Studies Ramón Margalef, Carretera San Vicent del Raspeig, 03690 Alicante, Spain
[4] Unit of Microbiology, Department of Biology and Geology, CITE II-B, Agrifood Campus of International Excellence CeiA3, CIAIMBITAL, University of Almeria, 04120 Almeria, Spain
* Correspondence: jose.saezt@umh.es

**Abstract:** Nowadays, plastic materials are extensively used in the agri-food sector for multiple purposes. The end-of-life management of these plastics is an environmental challenge because frequent incomplete recoveries after the crop seasons lead to the accumulation of plastics debris in agricultural waste, which is now recognized as an emerging environmental issue of global concern. However, the effects of plastic debris in agricultural waste undergoing biotreatment have been poorly studied. This study assesses the effects of agricultural plastic waste (APW) (LDPE + LLDPE and EPS) (1.25% f.w.) on the vermicomposting process (45 days) in terms of earthworm health by measuring biomarker responses and the enzymatic activity and quality/stabilization of the vermicompost obtained. The results showed that exposure to all the plastic materials tested had negative morphological effects on earthworm survival and body biomass. In the vermicomposting process, the changes detected in the enzymatic activity of the vermicompost and the biofilm seemed to affect the degradation rate of earthworms and the microbiome of the substrate, as demonstrated by the low organic matter mineralization in the vermicompost exposed to plastic. Although no significant changes were recorded in several biomarkers, signs of oxidative stress were evidenced throughout the glutathione S-transferase and carboxylesterase activity, mainly involving balanced oxidative stress and xenobiotic resistance systems.

**Keywords:** *Eisenia fetida*; vermicomposting; ecotoxicology; earthworm; agricultural plastic waste

## 1. Introduction

Different types of plastic materials widely used in agriculture have made the accumulation of plastic debris a global environmental concern. About twenty groups of plastics have been identified for agricultural use, with different formulations and a wide range of additives, such as chemicals to enhance elasticity, rigidity, UV stability, flame retardation, and color [1]. Polyethylene-based polymers are the most common plastics used in agriculture [2] because of their low cost, good workability, high impact resistance, excellent chemical resistance, and electrical insulation properties. Two grades of polyethylene plastics are low-density polyethylene (LDPE) and linear low-density polyethylene (LLPDE), which are thermoplastics made from ethylene monomers. They are mainly used to produce films (for greenhouses, low tunnels, mulching, UV protection, and silage) due to their resistance to tearing and impact [3].

Since synthetic plastics are durable because of low biodegradability, they have accumulated rapidly in the terrestrial environment, producing detrimental effects. Several studies have reported that agricultural films account for 10 to 30% of all the microplastics (MPs) accumulated in agricultural soil [4,5]. In addition to the accumulation of plastic mulch residues in farmland soils, other sources of plastic debris may be municipal waste [6], sewage sludge [7], and organic agricultural waste and its derived compost [8]. Flooding or water runoff can also lead to plastic debris accumulation in water bodies [9]. Little is known about the specific amount of plastic waste found in bio-waste, but in a study on the sewage sludge produced in a wastewater plant in Europe, the sludge contained between 1000 and 24,000 mg kg$^{-1}$ of plastic debris [10].

Composting is an environmentally friendly and organic method of bio-waste management. However, the safe use of compost or vermicompost must be guaranteed. For this, a complete characterization of these materials must be carried out prior to their use by the determination of the main chemical parameters commonly used to evaluate the maturity and stability reached in the organic matter content, e.g., water soluble carbon or humic and fulvic acid compounds [11] or even biological parameters, such as enzymatic activities, which are the most suitable methods to assess changes in aerobic biological activities [12]. Historically, composting has been used to recycle agricultural waste, as well as many other organic wastes, and the composted organic matter is returned to the soil to maintain soil fertility and crop productivity with minimal synthetic chemical fertilizer use.

However, due to the accumulation of plastic debris in bio-waste, one of the main challenges facing the compost industry today is contamination with plastic waste. Although most of this plastic can be removed before and after composting by sieving and manual sorting, and biodegradable plastic might be degraded during composting, plastic is still commonly found in the final product. In previous studies [1,13], concentrations of visible plastic ranging from 2.38 mg to 1200 mg kg$^{-1}$ of compost were found in different types of compost from commercial composting plants. Consequently, compost must be seen as a serious entry route for plastic in soil. Such plastic inputs may be especially problematic in agricultural soil. A yearly application of 7 t ha$^{-1}$ to 35 t ha$^{-1}$ can lead to an annual plastic input of between 1.2 and 6.3 kg ha$^{-1}$ to arable fields.

Several authors have investigated the combined use of composting and vermicomposting to treat different organic materials, showing that prior composting can accelerate degradation and improve the stabilization of the final product [14–16]. In addition, the use of earthworms allows for the assessment of the potential hazardous characteristics of bio-waste, evaluating their ecotoxicological responses by analyzing several oxidative stress biomarkers (catalase, glutathione S-transferase, and thiobarbituric acid reactive substances), which are involved in antioxidant enzymatic activity changes and in the production of several molecules related to responses of their immunological system [17].

This added-value fertilizer in vermicomposting is often reflected in the price. In Spain, the mean value of commercial compost is 34 € t$^{-1}$, while vermicompost can reach 200 € t$^{-1}$. Therefore, vermicomposting represents a cost-effective biotreatment with low technical requirements to convert organic waste into stabilized humic-like products.

The effect of accumulated plastic waste has been widely studied in marine [18–21] and terrestrial ecosystems [22–24].

Plastics in soil adversely affect plant health and soil fertility [25], water holding capacity, and soil microbial activity [26]. Microplastics may also act as carriers of other pollutants such as heavy metals, increasing their bio-accessibility [27]. Furthermore, in a study on plastic material added to soil, the authors in [28] reported that plastic material might act as a microhabitat, being rapidly colonized by microorganisms that form a dense biofilm on the surface of the plastic, named the plastisphere. This plastisphere consists of several layers. In the closest layer to the plastic surface, known as the ecological corona, the reactivity of the plastic material is higher, and partial inhibition of microbial activity can be observed. Biofilm formation on non-compostable plastics, such as LDPE, polystyrene (PS), and polyethylene terephthalate (PET), has also been described in marine ecosystems [29].

However, no study has explored the formation of biofilm on plastics exposed to organic matrix substrates such as compost or vermicompost. These could act as potential bio-stimulants due to their inherent microbiome content and activity. Therefore, despite the growing concern about plastic accumulation in bio-waste from agro-industrial activities, little is known about the effect of plastics in bio-treatment processes, such as composting or vermicomposting, their induced effect on enzymatic and hydrolytic activity, or the agronomical quality of the bio-fertilizer obtained.

The main aim of this study is to assess the effect of the presence of different plastics used in agriculture (LDPE + LLDPE and EPS) on the vermicomposting of bio-waste at a lab scale. The following implications were studied: (1) the evolution of the vermicomposting process and quality and stabilization of the final vermicompost; (2) the response and health of *E. fetida* (EF) by measuring the main biomarkers related to oxidative and damage stress; and (3) the enzyme activity in vermicompost and plastic-biofilm provoked by earthworms interacting with plastic material.

## 2. Materials and Methods

### 2.1. Experimental Design

The experimental design consisted of a lab-scale bioassay where earthworms were exposed to different types of agricultural plastic waste (APW) in bio-waste under vermicomposting conditions. Three plastic materials commonly used in agriculture were selected: low-density polyethylene (LDPE) + linear low-density polyethylene (LLDPE) black film, LDPE + LLDPE perforated film, and expanded polystyrene (EPS). To determine the normal behavior of *E. fetida* in feedstock, three replicates ($n = 3$) were prepared as control treatments with only feedstock and earthworms and no plastic material added, three replicates with feedstock with plastic added and no earthworms as composting treatment, and three replicates with feedstock, plastic added, and earthworms as vermicomposting as treatment for each type of plastic material tested. The bioassay consisted of an incubation (45 days) period in Petri dishes (15 cm ø) with 80 g of feedstock adjusted with distilled water to 70% moisture content. Then, 1 g of plastic material was added per replicate (1.25% f.w. proportion) and inoculated with 25 citellated *E. fetida* adults to simulate the vermicomposting treatment. The incubation containers were kept in isolated chambers under controlled conditions ($20 \pm 2$ °C and darkness).

The dose of plastic material (1.25% f.w.) added to the compost for this study was selected following the guidelines of Regulation (EU) 2019/1009 of The European Council on the limit of impurities (2 mm) of glass, metal, or plastic in commercial compost. Therefore, to consider the remaining material as compost fertilizer, the APW in the microcosm assays was below this range.

### 2.2. Experimental Set-Up

#### 2.2.1. Feedstock Characteristics

In the exposure bioassay, compost made from agroindustrial waste was used as feedstock. This compost was developed in the COMPOLAB-UMH facility at a commercial-pile scale (10 m³) using three ingredients (agri-food sludge + cow manure + vineyard pruning, in proportions of 45 + 15 + 40 vol %, respectively). The initial mixture was done with these proportions in order to adjust the C/N ratio to values close to 25 to improve the development of the composting process. The composting process lasted 96 days, including four turning events. The composting was used as pre-treatment to remove compounds harmful to earthworms, such as ammonium, and was stopped when the material completed the thermophilic phase. High-quality standards were achieved in terms of stabilization, sanitization, and the absence of phytotoxic effects (Table 1). The heavy metals also complied with fertilizer regulations (Regulation (EU) 2019/1009).

**Table 1.** Characteristics of the feedstock used.

| | Physicochemical Parameters | | | | Macronutrients | | | | Mature Parameters | | |
|---|---|---|---|---|---|---|---|---|---|---|---|
| pH | EC (dS m$^{-1}$) | BD (g L$^{-1}$) | TOM (%) | TOC (g kg$^{-1}$) | TN (%) | P (%) | K (%) | GI (%) | $C_{HA}$ (%) | $C_{FA}$ (%) | CEC (meq 100 g$^{-1}$ MO) |
| 7.8 | 4.5 | 486 | 63.0 | 225 | 2.13 | 0.41 | 1.01 | 108 | 1.93 | 2.27 | 128 |

EC: Electrical conductivity, BD: Bulk density, OM: Organic matter, TOC: Total organic carbon, TN: Total nitrogen, GI: Germination index, $C_{HA}$: Acid humic-like carbon, $C_{FA}$: Acid fulvic-like carbon, CEC: Cation exchange capacity.

### 2.2.2. Earthworms

The earthworms used in this study were obtained from large rearing containers (0.5 m$^3$) kept under controlled conditions (20 ± 2 °C and darkness). The earthworms were fed on the same feedstock used in the lab bioassay for 30 days to improve their adaptability. Adult citellated earthworms with a body mass of between 250 mg and 600 mg were selected, as recommended by international guidelines [30]. When mortality was observed during exposure, the worms were immediately removed from the rearing container.

### 2.2.3. Plastic Material

Three different plastic materials commonly used in agriculture were tested. Two kinds of plastic film (black film and perforated film) purchased from SolPlast Company (Murcia, Spain) composed by a mixture of low-density polyethylene (LDPE) and linear low-density polyethylene (LLDPE), which are commonly used in mulching film, were used. The LDPE + LLDPE resins have better mechanical properties than LDPE, such as higher tensile strength and impact and puncture resistance. These features make them more resistant to biodegradation. We also tested expanded polystyrene (EPS), which is used in pots for seedlings in horticultural crops. Small, irregular-shaped pieces of approx. 1 cm$^2$ were cut with scissors. The EPS materials were cut into small pieces of approx. 1 cm$^2$ and 1–2 mm thick with a steel guillotine.

### 2.2.4. Biofilm Sample

As previously mentioned, plastic debris acts as a habitat, and it is colonized by microorganisms that form a dense biofilm on the plastic surface. To determine the behavior of the enzymatic activity in the biological corona of the biofilm, we took samples of this by carefully separating small pieces of plastic from the vermicompost/compost and scraping them with a spatula until the plastic pieces were clean, and the substrate attached was collected and treated the same as vermicompost/compost samples for the posterior enzymatic measures.

### 2.3. *Analytical Methods*

### 2.3.1. Vermicompost Physicochemical Parameters

After homogenization, each vermicompost sample was divided into two subsamples. One was used immediately to determine the moisture content, and the other subsample was frozen at −80 °C to monitor the enzyme activity, and the other was kept at 45 °C in an oven with forced aeration to dry. This subsample was then ground to obtain dust particles using an agate ball mill (RESTCH mod. MM400). The particles were then left to dry at 105 °C to further analyze physicochemical parameters.

The physicochemical parameters in the vermicompost samples were analyzed as follows: electrical conductivity (EC) and pH were measured in a 1:10 water extract (*w/v*); moisture content was determined after drying to a constant weight at 105 °C for 24 h; total organic matter (TOM) content was measured by loss on ignition at 430 °C for 24 h; and total organic carbon (TOC) and total nitrogen (TN) were determined by burning the samples at 1020 °C in an automatic elemental micro-analyzer (EuroVector Elemental Analyzer, Milano, Italy). After digestion ($HNO_3/H_2O$) (1:1, *v/v*) of dry samples in the microwave system (CEM, mod. MARS ONE), macronutrients such as P and K, among others (Ca, Cu, Mg, Fe, Mn, Zn), and toxic heavy metals (Cr, Ni, Cd, Hg, Pb) were measured by ICP-OES.

The humic-like content was measured in an extract with 0.1 M NaOH, from which fulvic acid-like C ($C_{FA}$) was separated through acid precipitation of the humic acid-like C ($C_{HA}$). The extracted ($C_{EXT}$) and supernatant (CFA) were analyzed in an automatic carbon analyzer for liquid samples (TOC-V CSN Analyzer, Shimadzu Company, Kyoto, Japan). The water-soluble carbon (WSC) was measured in a 1:20 water extract (*w/v*) using the same automatic analyzer for liquid samples.

### 2.3.2. Vermicompost and Biofilm Enzymatic Activity

The vermicompost samples were homogenized by grinding the aggregates in a ceramic mortar and adding $H_2O$. The water suspension was at a ratio of 1:50 (*w/v*), namely, 1 g to 50 mL $H_2O$. Suspensions were carried out at the moment of preparation or maintained at 4–5 °C for a maximum of 3 days. The biofilm was carefully separated from the substrate and scraped with a spatula until the plastic was clean. Later, the sample was homogenized at a ratio of 1:10 (*w/v*), namely, 0.1 g to 10 mL $H_2O$.

Carboxylesterase activity (CbE) (EC 3.1.1.1.) was measured by pouring aliquots (100 μL) from the sample and adding 380 μL of Tris-HCl 0.1 M buffer (pH 7.0). The enzymatic reaction was initiated by adding 20 μL of 1-naphthyl butyrate substrate (1-NB) (2 mM, final concentration) and waiting 5 min before stopping the reaction. The formed product (1-naphthol) was revealed by adding 50 μL of Fast Red ITR salt to 0.1% (*w/v*), dissolved at 2.5% (*w/v*), and Triton X-100 at 2.5% (*v/v*). Finally, the absorbance of the naphthol-Fast ITR complex was measured at 450 nm using an Asys HiTech UVM340 microplate reader (Asys HiTech Gmbh, Eugendorf, Austria). Carboxylesterase activity was expressed as nmol $h^{-1} g^{-1}$ of dried substrate, determined by a calibration curve built for 1-naphthol. Control (without substrate) and blank samples (without vermicompost) were used to correct the background absorbance and non-enzymatic hydrolysis of the substrates, respectively.

Dehydrogenase (DHE) activity was measured by weighing 0.1 g of sample and adding 750 μL of Tris-HCl 0.1 M buffer (pH 7.0) + 1 mL INT. This was homogenized in a vortex and kept at 40 °C in a water bath for 1 h in darkness (samples were shaken every 20 min). The reaction was stopped by adding 2.5 mL of stop solution prepared as a mixture of N-N′ dimethyl and ethanol in a 1:1 (*v:v*) relation. Two random controls were prepared with 750 μL TRIS without INT. The plate spectrometer measurements were read at 450 nm.

To determine the catalase (CAT) (EC 1.11.1.6.) activity, 1 mL was collected from the 1:50 (*w/v*) aqueous suspension and dispensed with 125 μL of hydrogen peroxide ($H_2O_2$). It was put in a rotor for 10 min to allow the reaction to take place and was then stopped with 125μL 3 M of sulfuric acid.

### 2.3.3. *Eisenia fetida* Survival and Body Weight

After 7, 21, 30, and 45 days of the exposure assay, the earthworms were gently extracted from the feedstock of each replicate (Petri dish) by hand. Then they were counted for survival, weighed in a precision scale, and this information was recorded. At the end of the microcosm bioassay (45 d), the worms in each replicate were sampled.

### 2.3.4. Earthworm Biomarkers

Six earthworms randomly selected from each test replicate were used for this analysis; the selected earthworms were previously depurated (24 h) in order to eliminate the organic substrate of the gut tract. The earthworms' bodies were homogenized in ice-cold buffer (pH = 7.4) made of 25 mM sucrose, 20 mM Tris-HCl buffer, and 1 mM EDTA by milling with potter (Heidolph Company). The homogenates were centrifuged at $9000 \times g$ for 20 min at 4 °C to obtain the post-mitochondrial fraction, which was aliquoted and stored at −80 °C until analysis.

The total protein content of *E. fetida* was determined in a 1:10 (*v:v*) aqueous dilution with bicichoninic acid (BCA). The reagent was heated at 60 °C for 15 min, and then read on the spectrometer at 630 nm. Acetylcholinesterase (EC 3.1.1.7) activity was spectrophotometrically determined in the presence of 3 mM acetylthiocholine iodide as substrate and 0.1 mM of DTNB (5.5′-dithiobis-2-dinitrobenzoic acid) by measuring the increased absorbance during the kinetic reaction, read at 412 nm. The enzymatic reaction rate was quantified against a blank without substrate for each measurement. To subtract the spontaneous hydrolysis of the substrate, a second blank was performed without the sample. Acetylcholinesterase was expressed as nmol min$^{-1}$ mg$^{-1}$ protein.

To determine CbE, 100 μL of homogenized tissue was added to 380 μL 0.1 M Tris-HCl buffer (pH = 8.4) and 40 μL 1-naphthyl butyrate (1-NB) 20 mM. The tubes were incubated at 20 °C for 10 min and then centrifuged for 5 min at 10,000 rpm. Then 150 μL of supernatant was transferred to new microplates, and the formation of 1-naphthol was revealed after adding 50 μL of a solution containing 0.1% Fast Red ITR. The microplates were stored in darkness for 20 min, and then the absorbance of the naphthol–Fast Red ITR complex was read at 450 nm.

Lipid peroxidation was measured in 50 μL of homogenized tissue added to 450 μL of reactive acid 2-thiobarbituric (TBAR) and butylhydroxytoluene (BHT). The reaction was maintained for 30 min at 90 °C. After that, 250 μL was dispensed in 96 deep-well microplates and read at 492 nm in a spectrometer. Enzyme activity was expressed as μg MDA mg protein. Three randomized samples were carried out without TBAR.

The glutathione-dependent antioxidant enzymes glutathione reductase (GR) (EC 1.6.4.2) and glutathione S-transferase (GST) were measured using the method described by [31,32], respectively. GR activity was determined in an aliquot of 50 μL of homogenized *E. fetida* body tissue in a reaction medium of 100 mM Na-phosphate buffer adjusted to pH 7.5, 1 mM oxidized glutathione (GSSG), and 60 μM NADPH. The kinetic reaction was measured at 340 nm in the spectrophotometer to determine the rate of NADPH oxidation. Specific enzyme activity was calculated using the extinction coefficient of 6.22 M$^{-1}$ cm$^{-1}$. Glutathione S-transferase was measured in a reaction mixture containing 100 mM Na-phosphate buffer adjusted to pH 6.5, 2 mM CDNB (1-chloro-2,4-dinitrobenzene), 5 mM reduced glutathione (GSH), and 30 μL of sample. The extinction coefficient of 9.6 mM$^{-1}$ cm$^{-1}$ was used to express the specific enzyme activity.

### 2.3.5. Statistical Analysis

The IBM SPSS Statics V.28 software package was used for the statistical analyses. To assess the significant differences in the results measuring survival, weight variation, exoenzyme activity in the vermicompost, and biomarkers, the multivariate general linear model (GLM) was used, considering the effect of t main variables (EF presence, plastic format, APW presence). LSD tests were also conducted with Tukey-b and DMS as post hoc tests.

We used factorial analysis of variance to determine the statistically-significant differences between EF presence, type of APW, and the interaction between these two factors. When the differences were significant, one-way analysis of variance (ANOVA) and the least significant difference (LSD) were conducted to establish the significant differences between means. Normal distribution and variance homogeneity were checked using the Shapiro–Wilk and Levene tests, respectively, before ANOVA.

## 3. Results

### *3.1. Effect of AWP on Vermicompost Physicochemical Parameters*

The pH values at the end of the bioassay remained in a suitable range for earthworm and microorganism activity (5.5–8.5) [33] in all the treatments. As shown in Table 2, significant differences were observed in the vermicomposting process with earthworms and without earthworms. In general, the pH values significantly increased in containers without earthworms and with plastic materials. The EC values in the vermicompost also showed significant differences. In the presence of APW material, the samples with earthworms showed the highest final EC values compared to the samples without earthworms, regardless of the type of plastic material tested. No significant difference was found in organic matter content with the presence of earthworms. All the treatments showed a decrease in total organic matter at the end of the bioassay compared to the initial feedstock. Slight differences were found among the plastic materials tested. The vermicompost samples exposed to EPS had the highest values of TOM at the end of the bioassay. TOC content showed a general sharp decrease in all the treatments when compared to the initial feedstock. Nitrogen increased significantly in all the treatments, regardless of the type of plastic material. As can be seen in Table 2, when comparing the results by types of plastic, the vermicompost sample with earthworms had the lowest TN values. The WSC values showed significant differences for both *Eisenia fetida* presence and type of APW plastic. The sample with *E. fetida* presence had lower values of WSC than that without the earthworm (Table 2). The concentration of humic acid compounds at the end of the bioassay showed a slight significant difference when comparing the treatments with earthworms and those without earthworms. Additionally, the results did not show any differences between the plastic treatments and the control treatment, except for LDPE + LLDPE black film, whose samples, both with earthworms and without earthworms, had the highest values of humic acid compounds.

**Table 2.** Evolution of the main physicochemical characteristics of compost/vermicompost.

| *E. fetida* Presence | Type of APW | pH | EC (dS m$^{-1}$) | TOM (%) | TOC (%) | TN (%) | P (%) | K (%) | WSC (g kg$^{-1}$) | $C_{FA}$ (%) | $C_{HA}$ (%) |
|---|---|---|---|---|---|---|---|---|---|---|---|
| Yes | No plastic t = 45 d | 7.42 a | 5.42 | 52.9 | 23.7 b | 2.24 a | 0.81 | 1.28 | 8.83 a | 3.66 c | 3.58 |
| | LDPE + LLDPE black film | 7.24 a | 4.81 | 53.6 | 26.2 c | 2.31 ab | 0.82 | 1.27 | 9.00 a | 2.63 a | 4.11 |
| | LDPE + LLDPE perforated film | 7.46 a | 4.42 | 55.7 | 24.4 b | 2.29 ab | 0.82 | 1.16 | 8.36 a | 2.47 a | 3.27 |
| | EPS seedling | 7.21 a | 4.56 | 57.1 | 24.6 b | 2.24 a | 0.81 | 1.19 | 8.33 a | 2.45 a | 3.17 |
| No | No plastic t = 45 days | 7.88 b | 4.17 | 55.3 | 27.3 c | 2.53 c | 0.72 | 1.36 | 9.89 b | 2.57 a | 3.16 |
| | LDPE + LLDPE black film | 8.00 bc | 3.26 | 52.4 | 24.3 b | 2.38 b | 0.83 | 1.08 | 10.3 b | 3.31 b | 4.96 |
| | LDPE + LLDPE perforated film | 8.20 c | 3.20 | 54.4 | 24.0 b | 2.34 b | 0.81 | 1.12 | 9.23 b | 3.01 b | 3.76 |
| | EPS seedling | 8.20 c | 2.90 | 55.2 | 22.6 a | 2.25 a | 0.81 | 1.11 | 10.2 b | 2.71 ab | 3.87 |
| **Main effects** | | | | | | | | | | | |
| *E. fetida* presence | Yes | 7.35 a | 4.92 b | 54.4 | 24.8 b | 2.26 a | 0.81 | 1.22 | 8.41 a | 3.01 b | 3.52 a |
| | No | 8.07 b | 3.38 a | 54.3 | 24.4 a | 2.37 b | 0.79 | 1.19 | 9.82 b | 2.86 a | 3.89 b |
| Type of APW | No plastic t = 45 d | 7.57 a | 5.0 c | 53.7 ab | 24.9 b | 2.33 b | 0.78 | 1.28 | 8.83 | 3.32 c | 3.42 a |
| | LDPE + LLDPE black film | 7.61 a | 4.04 b | 53.0 a | 25.3 b | 2.34 b | 0.82 | 1.18 | 9.65 | 2.97 b | 4.53 b |
| | LDPE + LLDPE perforated film | 7.83 c | 3.81 a | 55.0 ab | 24.2 a | 2.31 b | 0.81 | 1.14 | 8.80 | 2.74 a | 3.52 a |
| | EPS seedling | 7.70 b | 3.73 a | 56.1 b | 23.6 a | 2.24 a | 0.81 | 1.16 | 9.26 | 2.58 a | 3.58 a |

**Table 2.** *Cont.*

| *E. fetida* Presence | Type of APW | pH | EC (dS m$^{-1}$) | TOM (%) | TOC (%) | TN (%) | P (%) | K (%) | WSC (g kg$^{-1}$) | C$_{FA}$ (%) | C$_{HA}$ (%) |
|---|---|---|---|---|---|---|---|---|---|---|---|
| | | | | | Statistical significance | | | | | | |
| | *E. fetida* presence | *** | *** | ns | *** | ** | ns | ns | *** | *** | * |
| | Type of APW | *** | *** | * | *** | *** | ns | ns | ns | *** | ** |
| | *E. fetida* x APW | *** | ns | ns | ** | *** | ns | ns | *** | *** | ns |

EC: Electrical conductivity, TOC: Total organic carbon, OM: Organic matter, Cw: Carbon water-soluble. ns, *, **, *** indicate not significant, statically significant at $p \leq 0.05$, $p \leq 0.01$, $p \leq 0.001$, respectively. Average values ($n = 3$) in a column followed by the same letter are not significantly different at $p < 0.05$ (Tukeys and DMS test).

Regarding the heavy metal content in the vermicompost, although concentrations of Cu, Zn, Cd, and Co slightly increased after the vermicomposting process, probably due to the organic matter degradation and subsequent reduce of volume, the final levels met the European Union Eco-label requirements for ecological production. Therefore, they can be used as organic amendments in agriculture (Regulation (EU) 2019/1009). The concentrations of Cr and Ni were below detection limits (<0.01 mg kg$^{-1}$).

Dehydrogenase activity (DHE) is commonly used to measure overall microbial activity, since it is involved in the respiration chain of all microorganisms [34]. The DHE/WSC ratio links microbial activity with the amount of easily metabolized organic matter. In our study, the initial feedstock showed a high amount of WSC (18.9 g/kg), which could lead to a quick increase in degradative and hydrolytic activity by the microorganisms in feedstock and the gut microbiome of earthworms (Figure 1). In all the treatments, the vermicompost WSC decreased at the end of the bioassay, suggesting substrate depletion and indicating the correct evolution of microbial activity and the biotransformation of the available organic matter into more stable molecules. The DHE/WSC ratio showed remarkable differences depending on the factors used in the statistical analysis, namely, type of plastic and earthworm presence. The LLDPE + LDPE black film and LLDPE + LDPE perforated film without earthworms had the highest values for that parameter (8.06 and 8.00, respectively) (Figure 1).

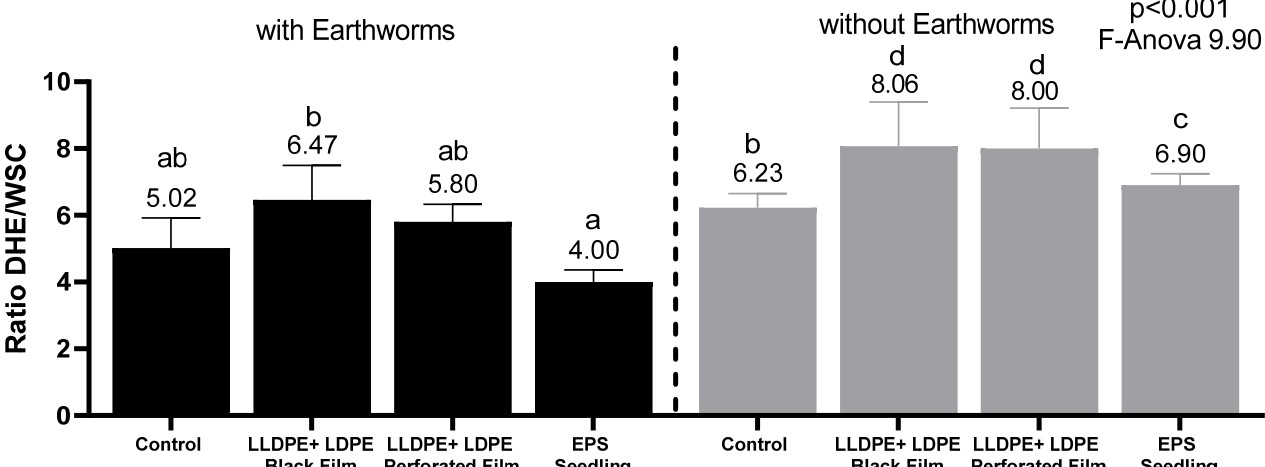

**Figure 1.** Graph of DHE/WSC ratio found in final vermicompost. Statical differences between treatments indicated by different letters at $p < 0.05$ (Tukeys and DMS test).

### 3.2. Vermicompost and Biofilm Exoenzymatic Activity

The enzymatic data obtained for the biofilm and vermicompost with different plastics are shown in Figure 2. The results for the vermicompost showed a significant difference in carboxylesterase (CbE) activity when compared to the control treatment without plastic. This behavior was observed in all the treatments, with and without earthworms (Figure 2a).

In all the cases under study, the treatments without EF presence (compost treatment) seemed to have more sensitivity to this CbE increase than vermicompost treatment with EF presence. In contrast, in the biofilm sample, the presence of EF appeared to promote an increase in the CbE enzyme compared to the biofilm sample without earthworms. Although a slight inhibition of the catalase enzyme was observed in the two kinds of plastic material tested (LDPE +LLDPE and EPS), the presence of plastic material did not significantly change the catalase activity compared to the control without plastic (Figure 2b). In the biofilm sample, the same behavior in all the treatments was observed. All the plastic treatments led to a sharp decrease in biofilm catalase activity compared to the substrate, with a mean decrease of 85% (Figure 2b). In our study, no significant differences among all the plastic treatments with earthworms and compost treatments without earthworms were shown by the ANOVA test for dehydrogenase activity (DHE) (Figure 2c). However, we highlight the increase in DHE activity observed in the compost treatment compared to the vermicompost, with 24.9, 37.3, and 53.3% increases for LDPE + LLDPE black film, LDPE + LLDPE perforated film, and EPS, respectively (Figure 2c). The biofilm did not show significant differences from the control treatments without earthworms.

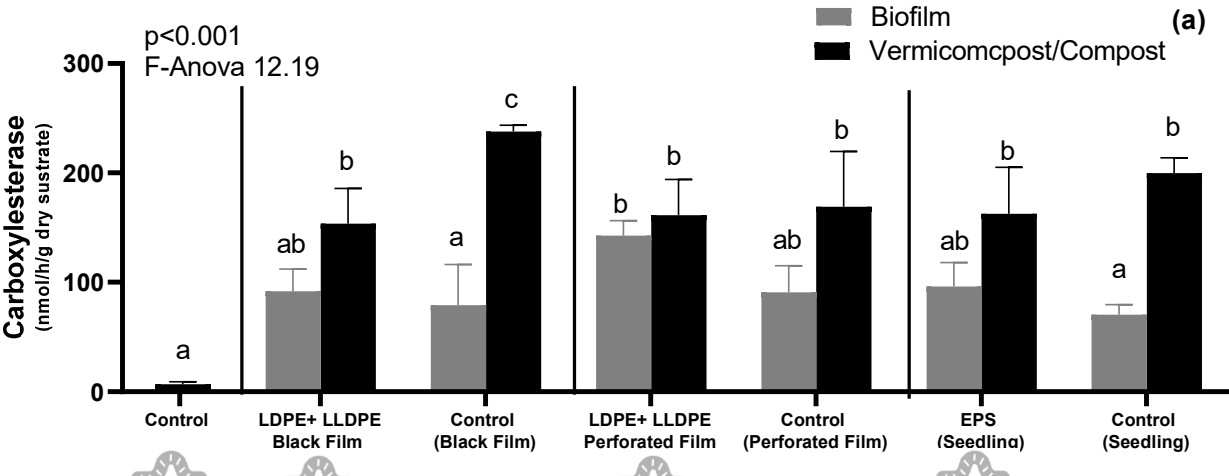

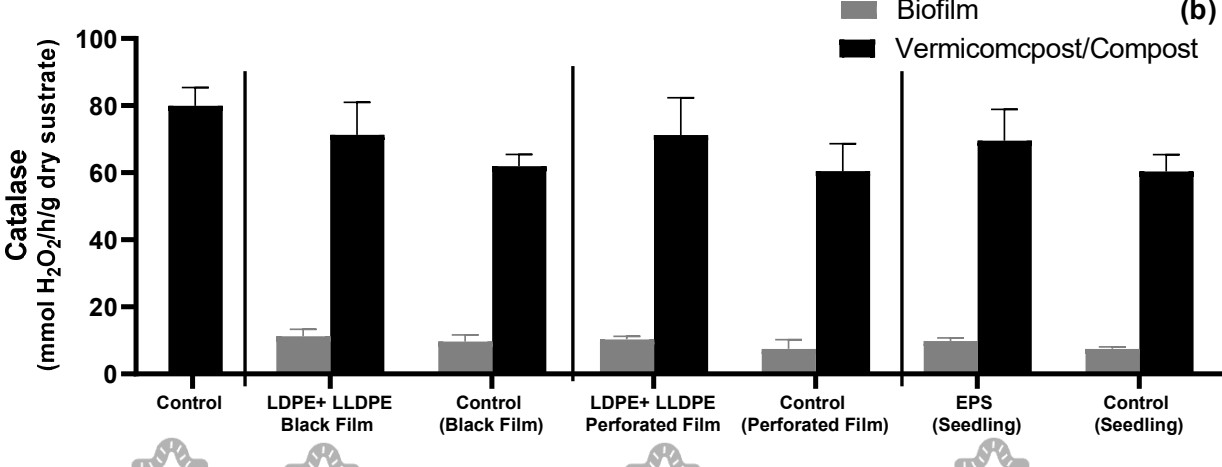

**Figure 2.** *Cont.*

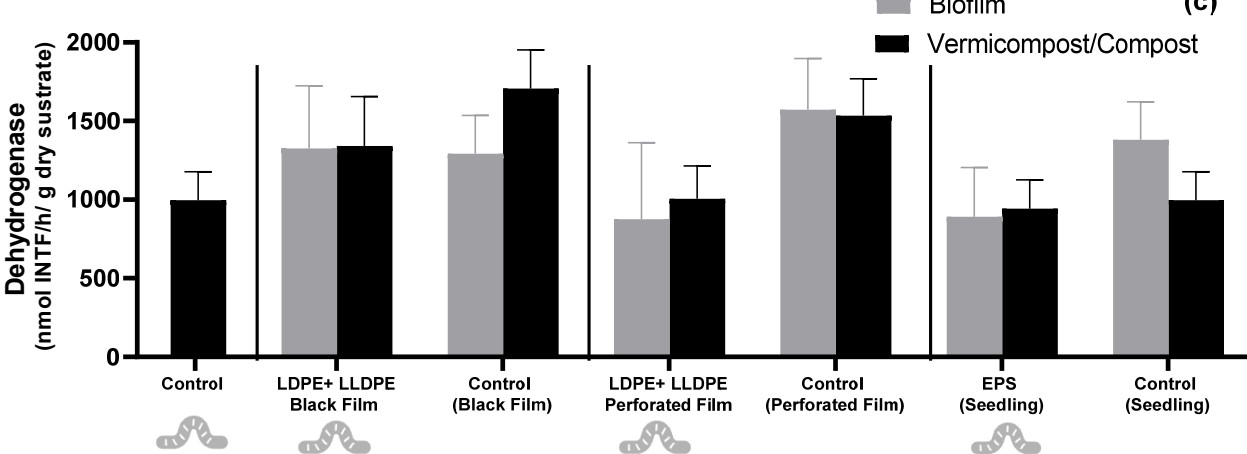

**Figure 2.** Graph of (**a**) carboxylesterase, (**b**) catalase, and (**c**) dehydrogenase activity determined in biofilm and vermicompost sample. The earthworm symbol indicates their presence in the vermicompost. Statical differences between treatments indicated by different letters at $p < 0.05$ (Tukeys and DMS test).

### 3.3. Eisenia fetida Survival and Body Weight

The control treatment maintained a higher density of earthworms with less mortality than the plastic treatments (Figure 3a). Therefore, the presence of APW seemed to decrease the survival rate of *E. fetida*. Significant effects were detected for the three different kinds of plastic tested. As shown in Figure 3a, earthworm mortality was observed mainly at the beginning of the microcosm bioassay. In the three plastic treatments, the rate of survival decreased in the initial stage, followed by stabilization until the end of the bioassay, except in the control treatment without plastic materials. At the end of the bioassay, the highest mortality was observed in the LLDPE + LDPE-black film treatment, with a decrease of 25% survival compared to the control (Figure 3a). Regarding the average body weight measured in all the specimens of each treatment, the control treatment enhanced *E. fetida* body weight compared to the plastic treatments, even though a higher density of earthworms was maintained with less mortality.

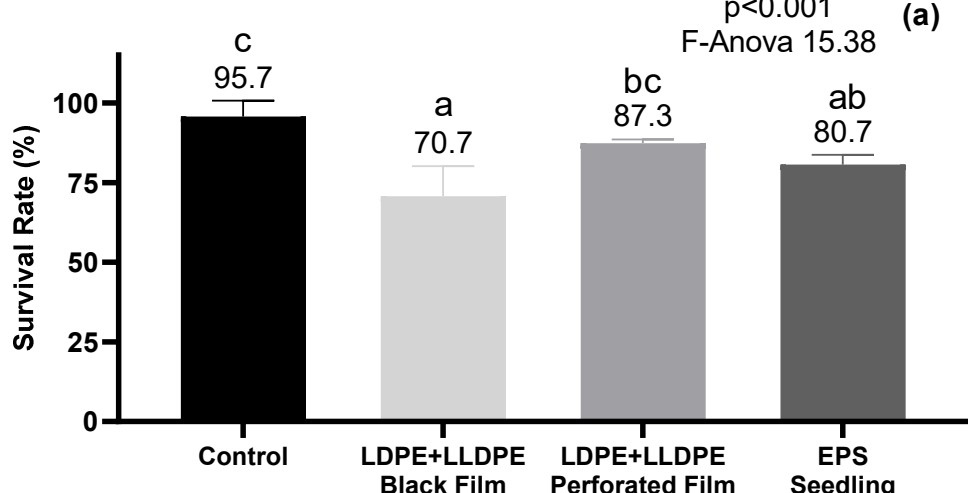

**Figure 3.** *Cont.*

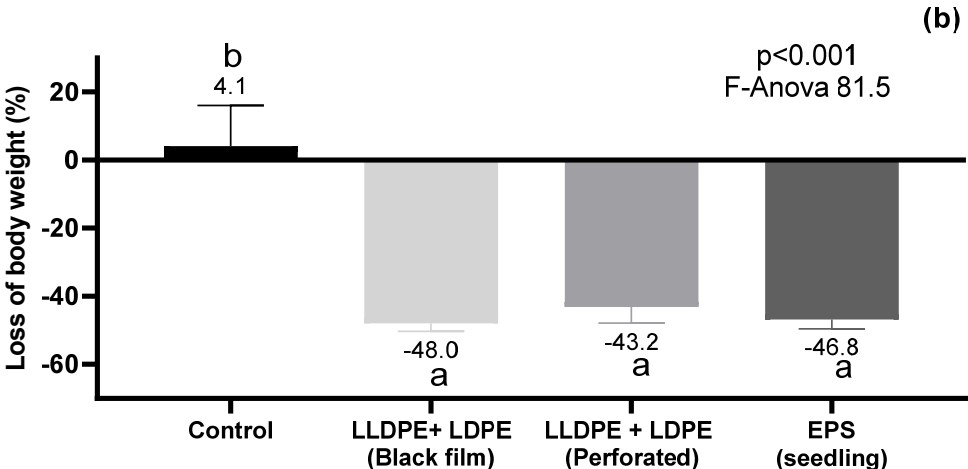

**Figure 3.** *Eisenia fetida* (**a**) survival and (**b**) loss of body weight. Statical differences between treatments indicated by different letters at *p* < 0.05 (Tukeys and DMS test).

### 3.4. Earthworm Biomarkers

Carboxylesterase is an esterase enzyme that plays a key role in the metabolic process of detoxification. It is considered an efficient protective mechanism for xenobiotic resistance in *Eisenia fetida* [35]. As shown in Figure 4a, a slight but insignificant increase in CbE activity was observed in all the treatments except EPS. Previous studies suggest that the luminal content of earthworms is the main source of CbE, which is released from the gut epithelium [36]. Due to the size and shape of the plastic tested in our study, the earthworms could not ingest the plastic. Regarding lipid peroxidation (Figure 4b), the results obtained are contrary to those expected, since we observed a decrease in lipid peroxidation. Our results indicate that the AChE in the *Eisenia fetida* exposed to LLDPE + LDPE and EPS were slightly affected compared to the control, although, as shown in the figure, not significantly (Figure 4c). The GST and GR activity in *Eisenia fetida* exposed to LLDPE + LDPE film plastic and EPS for 45 d was determined to characterize the effects of plastic on antioxidant defenses. In glutathione S-transferase, the response was a significant increase in earthworm body tissue activity after exposure to the plastic material (Figure 4d). The result of the glutathione reductase showed a significant change in earthworm exposure to LLDPE + LDPE in both the black film and perforated film, while EPS only caused a slight non-significant increase compared to the control treatment (Figure 4e).

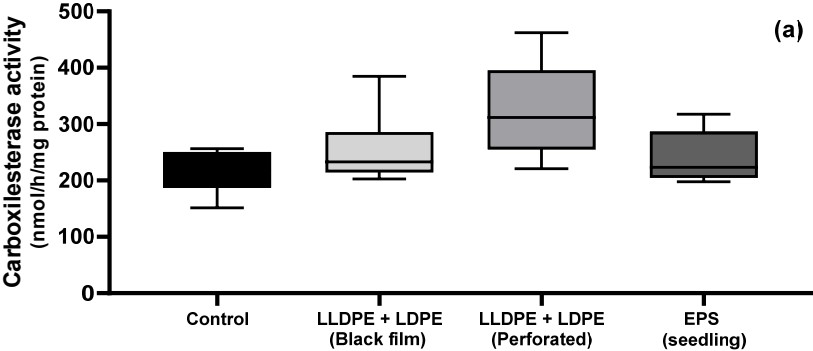

**Figure 4.** *Cont*.

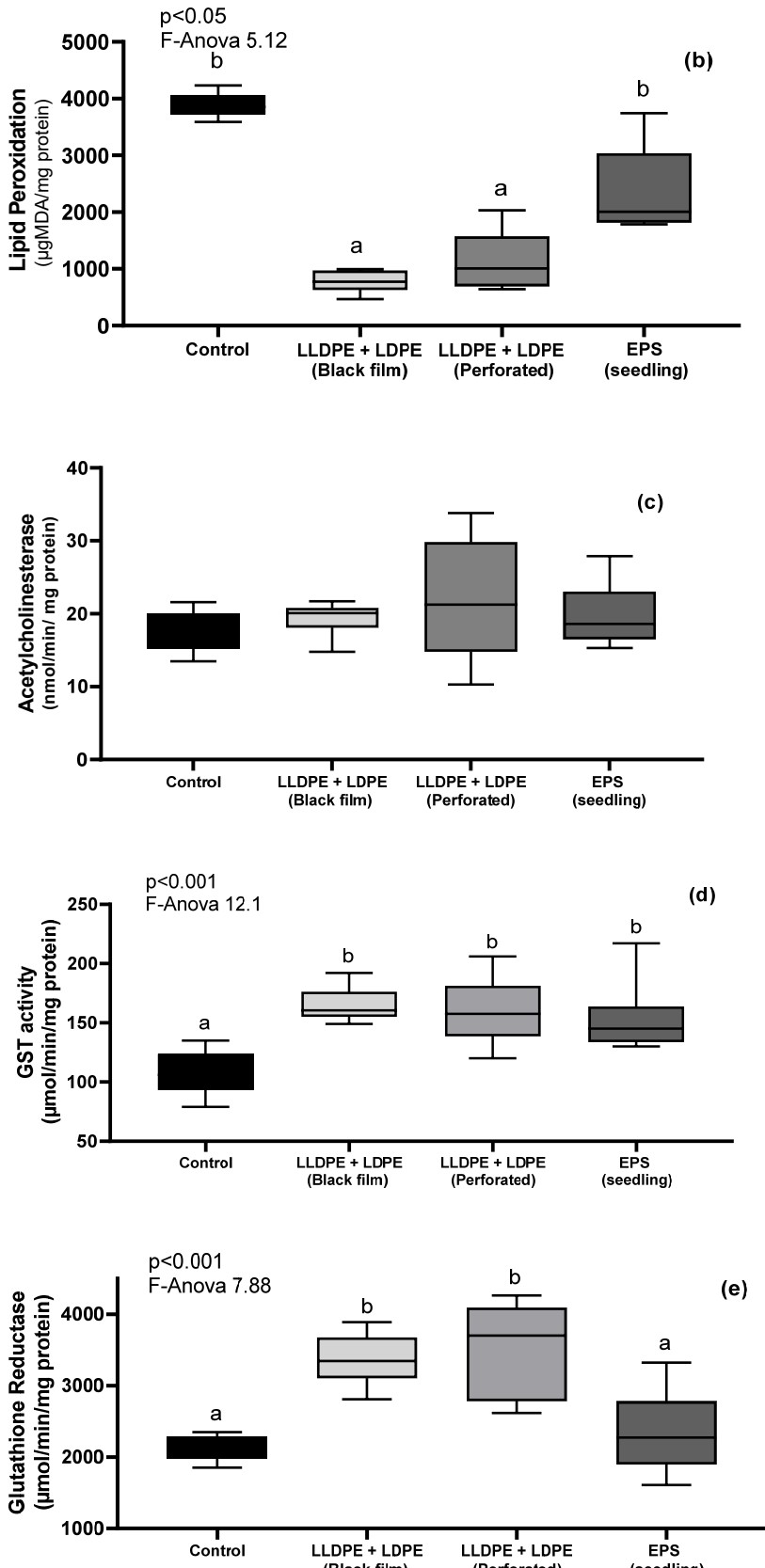

**Figure 4.** Carboxylesterase (**a**), lipid peroxidation (**b**), acetylcholinesterase (**c**), GST (**d**), glutathione reductase (**e**) activity in *E. fetida* body tissue. Statical differences between treatments indicated by different letters at *p* < 0.05 (Tukeys and DMS test).

## 4. Discussion

This study aims to gain knowledge about the effect of plastic presence in bio-waste during its treatment. The exposure bioassay of *Eisenia fetida* to APW presence was carried out with a concentration of plastic material (1.25% f.w.) that the European normative allows to be considered as compost fertilizer. The results obtained showed no significant effects on macronutrient NPK content. However, with plastic presence, significant changes were observed in other physicochemical characteristics of vermicompost such as WSC content or EC related to the degradation process. This affectation also was observed in vermicompost exposure to plastic waste, with an increase in CbE activity and remaining DHE activity. It is interesting to note the significant inhibition of CAT activity observed in biofilm samples, which we can hypothesize some organic plastic additives may have caused. In general, the result obtained seem to suggest a response of the microbiome to plastic exposure that leads to slowdown in the degradative process. Additionally, the earthworms showed negative morphological effects and mortality with the APW presence in feedstock. Additionally, the response of *E fetida* to plastic exposure show signs of oxidative stress, such as enhanced CbE activity or GST activity in body tissue induced by both types of plastic (EPS and LDPE + LLDPE).

An increase in pH values could be due to the microbiota, which utilizes the carbon fraction of the amino acids as an energy source and releases ammonia, causing an increase in pH (Table 2). In all the treatments with earthworms containing plastic, there were significant decreases in pH. Several reasons may explain this decrease: (1) the mucus from the *E. fetida* added to the ingested materials has been demonstrated to neutralize substrate [37]; (2) earthworms have shown excellent pH neutralization efficiency due to their calciferous glands [38]; and (3) the have an ability to regulate the release of organic acids depending on the characteristics of the starting feedstock [39]. pH values close to neutrality indicate the maturity of the vermicompost [40].

Increased EC in the vermicomposting process agrees with the findings by other authors [41,42] (Table 2). The reason for the rise in EC in the treatment with earthworms could be due to the higher mineralization of the organic matter, which released nutrient ions and soluble salts [43]. This is in contrast to the decrease in organic matter observed in the treatment without earthworms, but it could be explained by the ability of earthworms to promote some hydrolytic enzymes. These enzymes are not only linked to the C cycle (e.g., β-glucosidase), but also to N mineralization (e.g., urease) or the phosphorous cycle (phosphatase), which removes phosphate groups from organic matter [44]. The EC values in all the treatments exceeded the threshold of 4 dS m$^{-1}$ (Table 2), which is considered a limiting factor for plant cultivation [45]. However, EC values below 8 dS m$^{-1}$ are suitable for earthworm growth and development.

Earthworm mucus and activity are known to accelerate the rate of organic matter mineralization during the vermicomposting process, consequently leading to losses of total organic carbon by biotransformation [46,47]. The data obtained in this study did not show a great decline in organic carbon in the samples with earthworms. The final values of TOC content showed a significant difference between APW and earthworms, but no clear relationship between the presence of plastic and organic carbon evolution during the process was found. This drop in N in the sample with earthworms might be because parts of its initial content were transformed into body protein [48]. This behavior was also observed in the control treatment with and without earthworms, so it does not seem to be caused by exposure to plastic material. The mean values of total nitrogen (2.31%) were similar to those reported for other vermicompost (2.4%) made from agroindustrial waste, such as olive mill wastewater [31] (Table 2). The WSC content gradually decreased as vermicomposting progressed, in accordance with the consumption of available carbon sources for earthworm tissue formation and the subsequent stabilization of the substrate [30]. This behavior observed in LDPE + LLDPE film could be attributed to interactive mechanisms between this type of plastic material and the dissolved organic matter (DOM) content at a molecular level. Ref. [49] integrated spectroscopic methods into chemometric analyses, revealing the

microstructural exchange of DOM and plastic material, where polystyrene polymer-based plastic material interacted with the aromatic structure of DOM via a $\pi$–$\pi$ conjugation. DOM was then trapped in the plastic polymers by carboxyl groups and C=O bonds, with the subsequent increase in concentrations corresponding to humic-like substances under pH conditions ranging from 7 to 9.

In general, the plastic samples without earthworms showed higher values of the DEH/WSC ratio (Figure 1) than the group with earthworms. This may indicate higher metabolic potential remaining in the feedstock without earthworms. The control treatments without plastic and with plastic, both in vermicompost with earthworms and compost without earthworms, had statistically higher values of DHE/WSC at the end of the bioassay (Figure 1). Therefore, the results suggest that the earthworms accelerated the hydrolytic phase of the feedstock, while the plastic caused a slowdown in the degradative gut processes of the microbiota and inherent microbiota in the compost/vermicompost.

The CbE response could be due to the gut-associated processes of the earthworms (Figure 2), which could have alleviated the increase of CbE on the substrate but caused higher concentrations of CbE activity close to the surface of the film as a detoxification response. Found in polluted soil, carboxylesterase enzymes are effective exoenzymes with the capacity to degrade a wide range of organic compounds [50]. This esterase is even an efficient mechanism for deactivating organophosphorus pesticides, because the pesticide remains irreversibly bound to the active site of the enzyme [51]. A previous study [49] about sewage sludge vermicomposting reported significant inhibition in catalase activity in treatments with high heavy metal content. In other studies on soil pollutants, ref. [52] found clear catalase activity inhibition in soil treated with pesticides (chlorpyrifos), suggesting that the response was associated with a change in microbial activity (Figure 2). Ref. [53] reported a negative effect on salt-affected soil in biochemical processes with a sharp decline in catalase. A possible reason for this reduction at lower levels could be that some plastic additives acted as catalase inhibitors. Hydroxylamine is widely used in mulch as a UV and light stabilizer, while resorcinol is an efficient gas barrier in several polymers [22]. In all the cases, the gut-associated processes of earthworms increased the release of catalase enzymes in the biofilm (mean value 10.4 mmol $H_2O_2$ h $g^{-1}$ dry substrate), although not significantly compared to the control (mean value 8.1 mmol $H_2O_2$ h $g^{-1}$ dry substrate) (Figure 2).

Increased DHE activity in the compost treatment compared to the vermicompost (Figure 2) might indicate remaining high metabolic activity in the control treatments without earthworms. This increase in DHE could indicate a lower stabilization in the compost samples caused by the higher degradative rates induced by earthworm gut-associated processes in the vermicompost. In addition, a slight difference was found between the control and plastic treatments with earthworms, with mean values of 995, 1281, 1148, and 875 nmol INTF h$^{-1}$ g$^{-1}$ for the control, LDPE + LLDPE black film, LDPE + LLDPE perforated film, and EPS, respectively. It is possible that the earthworms of the control treatment (without affectation for plastic presence) were able to consume the most available organic substances with the subsequent reduction in DHE enzyme activity. This seems to indicate that of the plastic material tested in this study, the LDPE + LLDPE, affected the earthworms' degradative capacity. The DHE behavior observed in the biofilm was similar to that observed in the vermicompost sample. This could be because of some available organic matter and microbial activity remained in the substrate attached to the plastics (Figure 2).

The effects observed in epigenic earthworms when they are exposed to plastic material under vermicomposting conditions have been described as a set of biotic factors involving various physiological processes, such as respiration rate, reproduction rate, feeding rate, and burrowing activity [33] (Figure 3). Their body weight showed that the nutrient capacity of the feedstock material was not a limiting factor, since earthworms consume half of their weight per day [54]. Therefore, we can assume that the weight loss in the plastic treatments was caused by stress in the earthworms' physiological activity. The maximum body biomass of *E. fetida* was reached at 21 days of bioassay in the control test vessel. In

contrast, earthworm body weight constantly decreased in all the plastic treatments during the study. No significant difference was detected between plastic materials. Similar values of negative weight variation were obtained for LLDPE + LDPE-black film, LLDPE + LDPE perforated film, and EPS seedlings at the end of the bioassay. No reproduction or cocoon presence was observed during the duration of the bioassay.

Increased CbE activity in *E. fetida* could be explained by the fact that worms must produce greater amounts of the protein α, β-hydrolase, which promotes CbE, to catalyze the hydrolysis of some xenobiotic compounds released from plastic debris (Figure 4). Recent studies have demonstrated that microplastics with varied chemical compositions can cause skin damage, tissue lacerations, immunity disruption, and neurotoxicity in terrestrial organisms such as ciliates, collembolans, and earthworms [55,56]. Some studies have even shown that *E. fetida* ingestion of MPs (HDPE, PP, and LDPE) smaller than 300 μm [57,58] leads to inflammatory processes between the gut epithelium and the chloragogeneous tissue, sometimes with the development of fibrosis and congestion [59]. Our results showed that the control treatment reached the highest value at the end of the exposure bioassay. All the plastic treatments, except EPS, followed the same trend, with low levels of lipid peroxidation. Previous studies have reported that MP size can significantly influence toxicity [60]. Lei et al. [61] found that the adverse effects of MPs were closely related to their size rather than their composition in zebrafish (*Danio rerio*) and in nematode such as *Caenorhabditis elegans*. Thus, we speculate that the mortality induced in *E. fetida* due to LDPE + LLDPE film plastic exposure led to low specimen density, which allowed the *E. fetida* to avoid this exposure and subsequent tissue damage. The size, high elasticity, and low rigidity of the polyethylene films tested support this hypothesis. Another possible reason could be the increased GST activity shown by *E. fetida* body tissue exposed to plastic material. Since this is an important antioxidant enzyme, it can scavenge lipid peroxides, thus contributing to reducing cellular oxidative damage [62]. In contrast to our results, the authors in [57] reported an increase in AChE in *E. fetida* exposed 21 and 28 days to 1.0–1.5 g kg$^{-1}$ LDPE in soil, while the authors in [63,64] showed that AChE activity was inhibited in the dissected gut tissue of *Eriocheir sinesis* and *Pomatochistas microps* exposed to PS microplastics and PE, respectively. Refs. [65,66] also found a decrease in AChE. Ref. [65] indicated adverse effects in cholinergic neurotransmission and, thus, possibly in the nervous and neuromuscular functions of juvenile fish (common goby—*Pomatoschitas microps*) following exposure to polyethylene microplastic (1–5 μm). They observed a 42% inhibition in brain AChE. Ref. [66] reported AChE inhibition in *Eisenia andrei* exposed to polystyrene–HBCD in soil after 7 d of exposure, showing a recovery to normal values after 28 d of exposure. As studies that investigated the same type of plastic obtained different results, this might also indicate varying types of action depending on several factors, such as the type of plastic, but also concentration, shape, size, and the potential influence of additives. Therefore, the action mechanism of plastic material on AChE is still not clear, but we can assume that the concentration of plastic (1.25%), as well as the size or shape of the plastics tested seemed to affect the low acute toxicity of *E. fetida*. The same behavior shown in regarding GST activity was observed by [66] in *Eisenia andrei* exposed to polystyrene–HBCD and car tire abrasion plastic present in soil. Furthermore, they reported a time-dependent response with increased GST activity from day 7 to day 28 of exposure. On the other hand, ref. [67] reported significantly inhibited GST activity in *E. fetida* after exposure to HDPE and PP microplastics for 14 days. A similar decreasing trend for GST activity was observed in *E. fetida* when exposed to low-density polyethylene (LDPE) and PS MPs [68]. Other studies have also reported that exposure to plastic material can upregulate the level of reactive oxygen species (ROS), thereby perturbing the antioxidant system [69,70]. Therefore, the immune response against ROS is a mechanism that requires an action–response balance. Exposure to LDPE + LLDPE or EPS resulted in the accumulation of ROS, which then stimulated the biosynthesis of antioxidant enzymes. Once the excess of accumulated ROS overwhelms the antioxidant defense systems, the synthesis or structure of antioxidant enzymes can be easily influenced, resulting in a decrease in enzyme activity [71]. Another

reason for the differences found in GST activity is the production of malondialdehyde, also known as a thiobarbituric acid reactive agent, as a product of lipid peroxides. This carbonyl compound is one of the most abundant end-products of lipid peroxidation and may have induced the GST activity through its elimination by conjugation with GSH [72]. The glutathione-dependent enzyme is related to a mechanism of ROS–GSH balance. These increases in GR activity could be caused by an antioxidant response.

## 5. Conclusions

The findings of this study suggest that the presence of LDPE + LLDPE and EPS (1.25% f.w.) in bio-waste allows their bio-treatment throughout composting or vermicomposting. However, signs of degradative process slowdown were observed in the enzymes measured, which can lead to a retardation of the hydrolytic phase. Nevertheless, the final characteristics of the vermicompost exposed to plastic did not show significant differences from the control vermicompost. Two types of plastic tested had a negative morphological effect and even mortality on *E. fetida*. The measured biomarkers reflected an antioxidant response through enhanced GST activity and a detoxification process through increased CbE in earthworm tissues. These results have extended our knowledge about the effects of agricultural plastic waste on bio-waste treatment. However, further future studies should include a wide variety of plastic types, concentrations, sizes, and shapes to better understand the mechanisms involved in oxidative stress.

**Author Contributions:** Conceptualization, R.M.; methodology, R.M. and M.J.L.; software, J.A.S.; validation, F.C.M.-E. and M.J.L.; formal analysis, J.A.S., Z.E.B.M. and A.M.P.T.; investigation E.M.-S.; resources, F.J.A.-R.; data curation, R.M.; writing—original draft preparation, J.A.S., Z.E.B.M. and A.M.P.T.; writing—review and editing, F.C.M.-E. and F.S.-E.; visualization, R.M. and F.J.A.-R.; supervision, M.J.L. and R.M.; project administration, M.J.L. and R.M.; funding acquisition, M.J.L. and R.M. All authors have read and agreed to the published version of the manuscript.

**Funding:** This research has received funding from the Bio-Based Industries Joint Undertaking (JU) under the European Union's Horizon 2020 Research and Innovation Programme under grant agreement No. 887648—RECOVER project. The JU receives support from the European Union's Horizon 2020 Research and Innovation Programme and the Bio-Based Industries Consortium.

**Conflicts of Interest:** The authors declare no conflict of interest.

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
