# Peer review of "The Effects of Agricultural Plastic Waste on the Vermicompost Process and Health Status of Eisenia fetida"

_agronomy, doi:10.3390/agronomy12102547_

Round 1

Reviewer 1 Report

1.You can improve the topic to make it more specific.

2.Lack of introduction of the research of point 1 and piont 2.

3.How to process the data of dead earthworms, how to count or weigh them?

4.Why choose 45d as the end time of the experiment? not any other day?

5.Why do heavy metal content in the vermicompost increase? if it necessary to provide data and analyze the reasons.

6.Because the effect of vermicompost mainly depends on the life of earthworms, in this study, how to explain the influence of more than 20% earthworm death factors on the composting results, or how to ignore the influence?

7.As the carbon nitrogen ratio is an important indicator of the maturity of vermicompost, it is suggested to pay attention to the change of carbon nitrogen ratio before and after the test, analyze the relationship with the experimental results with software.

Author Response

Response to Reviewer 1 Comments

Point 1. You can improve the topic to make it more specific.

Response 1: We want to thank the comments of the reviewer. The topic has been changed following the suggestion of the reviewer by:

“The effect of agricultural plastic waste on the vermicomposting process and health status of Eisenia fetida

We think that this topic is more specific for our manuscript.

Point 2. Lack of introduction of the research of point 1 and point 2.

Response 2: We have included a paragraph in the Introduction section completing the points mentioned by the reviewer and we have added their corresponding references.

Point 3. How to process the data of dead earthworms, how to count or weigh them?

Response 3. The earthworms are carefully removed manually from the substrate, counted and weighed for each replicate on a precision balance. We have tried to clarify this point better in the manuscript

Point 4. Why choose 45d as the end time of the experiment? not any other day?

Response 4. The decision to carry out the experiment in 45 days was an arbitrary choice. The reproduction cycle of earthworms is stablished in 28 days, but it seems a very short time for a vermicomposting process.

Point 5. Why do heavy metal content in the vermicompost increase? if it necessary to provide data and analyze the reasons.

Response 5. We appreciate this remark from reviewer. It is common that heavy metal contents slightly increase in composting/vermicomposting process due to the degradation of organic matter and the consequent loss of substrate volume. The increase observed in this experiment is very low (not statically significance) and the resulting vermicompost can be classified in the most restrictive category for this parameter (Class A) according with European normative.

Point 6. Because the effect of vermicompost mainly depends on the life of earthworms, in this study, how to explain the influence of more than 20% earthworm death factors on the composting results, or how to ignore the influence?

Response 6: We understand that the presence of plastic affected the earthworms and their metabolic system, and therefore to the characteristics of vermicompost obtained. Since the individuals exposed to the plastic material loss weight during the Bioassay, even with lower population density. While in Control treatment the individuals with more population density increase their body weight. Earthworm growth is density-dependent, and individual growth and earthworm weight are lower in crowded conditions.

Reference

Dominguez, J. (2018). “Earthworms- The Ecological Engineers of Soil” Chapter 5-Earthworms and vermicomposting. Science-Book (http://dx.doi.org/10.5772/intechopen.76088).

Point 7. As the carbon nitrogen ratio is an important indicator of the maturity of vermicompost, it is suggested to pay attention to the change of carbon nitrogen ratio before and after the test, analyze the relationship with the experimental results with software.

Response 7: We completely agree with the reviewer; CN ratio is an important parameter in the composting and vermicomposting processes. However, we have not included because we did not find significant or relevant differences in this parameter before and after the test; so, in this case the information provided by this parameter was not relevant for our study and we preferred to select other parameters with a higher influence.

Reviewer 2 Report

Issues concerning the accumulation of plastics and microplastics in the environment is a very current topic. Often, due to their low biodegradability, they are treated as inert substances, accumulating but having no impact on the surrounding biotopes. The authors showed that it affect the formation of vermicompost, which is a soil conditioner. The information verified by the authors can also be a valuable indication drawing attention to the importance of the process of separating plastic debris before the process of organic waste composting.

The article is presented in a concise form, with a short introduction covering the issue of the presence of plastic waste in the environment, and its impact on the environment but also the vermicomposting process. 

The research methodology is very clearly described.

Why did the authors decide to use cattle manure and vineyard prunings, in a combination of agri-food sludge + cow manure + vineyard prun, and in such a 45:15:40 ratio?

How was the processing similar to natural aging in the field of plastic? 

Figure 2 legend does not explain/specify the earthworm symbol used in the diagrams.

The first paragraph of the discussion looks like an excerpt from the authors' instructions. 

In my opinion, the element of the publication that needs improvement is the discussion.

The discussion need a major rewrite and a linguistic check. It begins abruptly, without any introductory sentence or general description of the results against the background of the literature and other studies. This is clearly visible in paragraphs that begin in lines  402, 452, 497.
It is written in a noticeably different style from the rest of the publication. The sentence constructions used are less professional and the sentences used are repetitive, e.g. Line 473, 478, 482 "This could be..."

The discussion requires rewriting and gathering the thoughts presented in the following sentences into a coherent whole accessible to the reader. As it stands, it looks in several places like a collection of notes created for the purpose of writing it. 

The conclusions are formulated correctly and in a concise form. 

Author Response

Response to Reviewer 2 Comments

Issues concerning the accumulation of plastics and microplastics in the environment is a very current topic. Often, due to their low biodegradability, they are treated as inert substances, accumulating but having no impact on the surrounding biotopes. The authors showed that it affect the formation of vermicompost, which is a soil conditioner. The information verified by the authors can also be a valuable indication drawing attention to the importance of the process of separating plastic debris before the process of organic waste composting.

The article is presented in a concise form, with a short introduction covering the issue of the presence of plastic waste in the environment, and its impact on the environment but also the vermicomposting process.

The research methodology is very clearly described.

Point 1. Why did the authors decide to use cattle manure and vineyard prunings, in a combination of agri-food sludge + cow manure + vineyard prun, and in such a 45:15:40 ratio?

Response 1. We want to thank the comments of the reviewer. Each raw material was previously characterized and the mixture was done is this proportions in order to adjust the C/N ratio considering the values in the initial materials, to improve the composting process. This point has been clarified in the manuscript.

Point 2. How was the processing similar to natural aging in the field of plastic?

Response 2. This is a mistake, sorry for it. The sentence has been eliminated.

Point 3. Figure 2 legend does not explain/specify the earthworm symbol used in the diagrams.

Response 3. The description of the symbol of earthworms has been included following the comment of the reviewer.

Point 4. The first paragraph of the discussion looks like an excerpt from the authors' instructions.

Response 4. This was a mistake, because this paragraph was part of the template manuscript of the journal, we apologise for this. The paragraph has been eliminated.

In my opinion, the element of the publication that needs improvement is the discussion.

Point 5. The discussion need a major rewrite and a linguistic check. It begins abruptly, without any introductory sentence or general description of the results against the background of the literature and other studies. This is clearly visible in paragraphs that begin in lines  402, 452, 497.

It is written in a noticeably different style from the rest of the publication. The sentence constructions used are less professional and the sentences used are repetitive, e.g. Line 473, 478, 482 "This could be..." The discussion requires rewriting and gathering the thoughts presented in the following sentences into a coherent whole accessible to the reader. As it stands, it looks in several places like a collection of notes created for the purpose of writing it.

Response 5. We appreciate this comment from the reviewer. We have rewritten the Discussion section to make it clearer and not repetitive. An introductory paragraph has been included as an overview of the results obtained. In addition, the manuscript has been revised by a native English-speaking person, Laura Wettersten.

Point 6. The conclusions are formulated correctly and in a concise form.

Response 6. We appreciate the comment of the reviewer.

Reviewer 3 Report

The end-of-life management of this plastics is an environmental challenge. The study investigated the effects of agricultural plastic waste on the vermicompost, and it is of importance to the compost. However, two problems need to be solved in the sutdy.

The expriement were conducted in the Petri dishes. Is it too simple for a compost process? Please add relevant references.

The whole experiment ran for 45 days, how about the data after 7, 21, 30 days of exposure?

Author Response

Response to Reviewer 3 comments

The end-of-life management of this plastics is an environmental challenge. The study investigated the effects of agricultural plastic waste on the vermicompost, and it is of importance to the compost. However, two problems need to be solved in the sutdy.

Point 1. The expriement were conducted in the Petri dishes. Is it too simple for a compost process? Please add relevant references.

Response 1:  We agree with reviewer in the sense that Petri dishes is a too simple system for simulate a composting process, due to the great microbial activity and the exothermic behaviour of the process. However, the purpose of this study was to evaluate the vermicomposting conditions at a mesocosms scale as a preliminary study. The use of Petri dishes as experimental environment for earthworms has been previously considered as a suitable medium for holding individual specimens or small groups of earthworms, since plastic Petri dishes allow gas exchange while also maintaining good moisture conditions in the substrate (Dominguez, 2018).

Reference

Dominguez, J. (2018). “Earthworms- The Ecological Engineers of Soil” Chapter 5-Earthworms and vermicomposting. Science-Book (http://dx.doi.org/10.5772/intechopen.76088).

Point 2.  The whole experiment ran for 45 days, how about the data after 7, 21, 30 days of exposure?

Response 2.  At the end of bioassay (45 days) were determined the physico-chemical parameters, enzymatic activity of vermicompost and biomarkers of Eisenia fetida tissue. Only the survival and weight variation were determined at 7,14,21 and 30 days, but we presented the final result (45d) in the manuscript because we considered them as the most relevant results.